# Are Cirrhotic Patients Receiving Invasive Mechanical Ventilation at Risk of Abundant Microaspiration

**DOI:** 10.3390/jcm11205994

**Published:** 2022-10-11

**Authors:** Clementine Levy, Alexandre Gaudet, Emmanuelle Jaillette, Jean Reignier, Guillaume Lassailly, Malika Balduyck, Emeline Cailliau, Anahita Rouze, Saad Nseir

**Affiliations:** 1Department of Intensive Care Medicine, Critical Care Center, CHU Lille, 59000 Lille, France; 2U1019-UMR9017-CIIL-Centre d’Infection et d’Immunité de Lille, Institut Pasteur de Lille, CNRS, Inserm, CHU Lille, University Lille, 59000 Lille, France; 3Department of Intensive Care Medicine, CHU Nantes, 44000 Nantes, France; 4Department of Hepatology & Gastroenterology, CHU Lille, 59000 Lille, France; 5Department of Biochemistry, CHU Lille, 59000 Lille, France; 6Department of Biostatistics, CHU Lille, 59000 Lille, France; 7UMR 8576-UGSF-Unité de Glycobiologie Structurale et Fonctionnelle, CNRS, Inserm U1285, Université de Lille, 59000 Lille, France

**Keywords:** cirrhosis, ventilator-associated pneumonia, microaspiration

## Abstract

Previous studies have identified cirrhosis as a risk factor for ventilator-associated pneumonia (VAP). The aim of our study was to determine the relationship between cirrhosis and abundant gastric-content microaspiration in intubated critically ill patients. We performed a matched cohort study using data from three randomized controlled trials on abundant microaspiration in patients under mechanical ventilation. Each cirrhotic patient was matched with three to four controls for gender, age ± 5 years and simplified acute physiology score II (SAPS II) ± 5 points. Abundant microaspiration was defined by significant levels of pepsin and alpha-amylase in >30% of tracheal aspirates. All tracheal aspirates were collected for the first 48 h of the study period. The percentage of patients with abundant gastric-content microaspiration was the primary outcome. The abundant microaspiration of oropharyngeal secretions, VAP incidence, the duration of mechanical ventilation, length of intensive care unit (ICU) stay and mortality were the secondary outcomes. A. total of 39 cirrhotic patients were matched to 138 controls. The percentage of patients with abundant gastric-content microaspiration did not differ between the two groups (relative risk: 0.91 (95% CI: 0.75 to 1.10)). There was no significant difference between the two groups in terms of the abundant microaspiration of oropharyngeal secretions, VAP, the duration of mechanical ventilation, the length of ICU stay and mortality. Our results suggest that cirrhosis is not associated with abundant gastric-content microaspiration.

## 1. Introduction

Ventilator-associated pneumonia (VAP) is a common complication of mechanical ventilation.

This infection is associated with increased morbidity, the duration of mechanical ventilation, mortality and healthcare-associated costs [1,2]. Microaspiration is a key factor in the pathogenesis of VAP [3,4]. The microaspiration of gastric contents and oropharyngeal secretions is related to tracheal tube presence, which prevents the closure of the vocal cords and the protection of the lower respiratory tract. Tracheal colonization can progress to ventilator-associated tracheobronchitis and VAP when the quantity and virulence of bacteria are high and local and general host immunity are altered. Other risk factors for microaspiration are enteral nutrition, supine position, use of deep sedation and other patient’s related factors [3]. 

Cirrhosis is a healthcare burden with frequent acute decompensation. Cirrhotic patients represents 2–5% [5] of all patients in intensive care units (ICUs). Digestive hemorrhage and sepsis are the main causes leading to ICU hospitalization [6]. Mechanical ventilation is often required in this setting and can lead to the development of VAP. Respiratory failure and uncontrolled sepsis are the two major risk factors for mortality in cirrhotic patients, with mortality reported ten times higher in ventilated patients [7]. 

Previous small studies and animal models [8,9] have suggested an increased risk of microaspiration in patients with cirrhosis, mainly due to intra-abdominal hypertension and ascites resulting in gastric reflux and aspiration. To our knowledge, there are no available data on the relationship between abundant microaspiration and cirrhosis.

Therefore, we conducted this study to evaluate the relationship between cirrhosis and abundant gastric microaspiration in mechanically ventilated patients. We also aimed to determine the relationship between cirrhosis and the abundant microaspiration of oropharyngeal secretions, VAP incidence, the duration of mechanical ventilation, the length of ICU stay and mortality.

## 2. Materials and Methods

For this post-hoc analysis, we used data from three randomized controlled trials on microaspiration performed in adult critically ill patients receiving invasive mechanical ventilation for more than 48 h. The first study was a single-center study evaluating the role of the continuous control of tracheal cuff pressure on the incidence of microaspiration [10]. The second was a multicenter study that evaluated the role of tracheal cuff shape on the incidence of microaspiration [11]. The third was a multicenter study that evaluated the role of enteral or parenteral nutrition on microaspiration in patients with septic shock [12]. 


**
Inclusion criteria 
**


Patients included in the three trials were eligible, provided that tracheal aspirates were collected and analyzed for pepsin.


**
Standard care protocol
**


All patients were positioned in a semirecumbent position, according to guidelines, during mechanical ventilation. Oral care was performed using chlorhexidine 0, 1%. A subglottic secretion–drainage device was not used in these studies.

Tracheal cuff pressure was continuously monitored in the intervention arm of the Nosten study [10] and manually adjusted with a manometer in the control group of the Nosten trial and in the two other studies. The tracheal cuff shape was standard in all patients, except those randomized in the tapered-cuff arm of the BestCuff study [11].


**
Measurement of pepsin and alpha-amylase
**


After randomization, all tracheal aspirates were collected for 48 h for pepsin and alpha-amylase quantitative measurements. Tracheal aspirates were only performed when clinically relevant (abundant respiratory secretion, mucus plug, patients’ discomfort or bacteriological sampling). All tracheal aspirates were stored at −20 °C and sent to the Lille University Hospital central laboratory. All measurements were blindly performed using an enzyme-linked immunosorbent-assay (ELISA) technique for pepsin measurement and the difference between the total and pancreatic amylase activity for the alpha-amylase dosage.


**
Study population
**


Exposed patients were cirrhotic patients under mechanical ventilation. Each cirrhotic patient was matched with three to four patients without cirrhosis (unexposed patients). A matching procedure was performed based on the following criteria: age (±5 years), simplified acute physiology score II (SAPS II) (±5) and gender, using an optimal-matching algorithm without a replacement. 


**
Ethic approval
**


This study was approved by the local Institutional Review Board. In accordance with French law, and because of the retrospective observational design, written informed consent was not required. 


**
Definitions 
**


Cirrhosis was defined by liver biopsy or clinical, radiological and laboratory features compatible with cirrhosis [13]. 

VAP was defined according to guidelines by a new infiltrate upon chest X-ray, associated with at least two of the following: fever > 38 °C or hypothermia < 36 °C; purulent tracheal aspirates; and hyperleukocytosis > 10 G/L or leukopenia < 1.5 G/L [1]. Microbiological confirmation was required in all patients. Microbiological identification was performed using positive bronchoalveolar lavage (BAL) > 10^4^ CFU/mL or tracheal aspirates ≥ 10^5^ CFU/mL. VAP occurring between three and five days of mechanical ventilation was defined as early VAP. VAP occurring after day 5 of mechanical ventilation was defined as late-onset VAP. Only first VAP episodes were taken into account.

Abundant gastric-content microaspiration was defined by significant level of pepsin (>200 ng/mL) in >30% of tracheal aspirates per patient. Abundant oropharyngeal microaspiration was defined by a significant salivary amylase level (>1685 IU/mL) in >30% of tracheal aspirates per patient [14]. 


**
Outcomes 
**


Abundant gastric-content microaspiration was the primary outcome. The abundant microaspiration of oropharyngeal secretions, VAP incidence, the duration of mechanical ventilation, the length of ICU stay and mortality were the secondary outcomes.


**
Statistical analysis 
**


Categorical variables are presented as numbers (percentage). Quantitative variables are presented as mean ± standard deviation or median and interquartile range according to the normality of distribution. The normality of distributions was checked graphically and using the Shapiro–Wilk test. Baseline characteristics were described according to exposure status (cirrhotic versus non-cirrhotic patients), and the magnitude of the between-group differences was assessed by calculating the absolute standardized difference (ASD). An ASD >10% was interpreted as a meaningful difference [2].

Abundant gastric and oropharyngeal microaspiration and abundant alpha-amylase rates were compared between the two study groups using a generalized estimating equation (GEE) model (using a binomial distribution and log link function) to account for the matched design. Relative risks (RRs) with a 95% confidence interval (CI) for the cirrhotic versus non-cirrhotic patients were derived from the GEE model as effect sizes. The cumulative incidence of VAP (censored at 28 days) was estimated using the Kalbfleisch and Prentice method to consider extubation (dead or alive) as a competing event, and it was compared between the two study groups using a marginal Fine and Gray model for clustered data to account for the matched design [15]. The subhazard ratio (sHR) for cirrhotic versus non-cirrhotic patients was derived from this model as the effect size. The length of mechanical ventilation and ICU stay (both censored at day 28) were also compared between the two study groups using competing survival risk analyses. We estimated and compared cumulative incidences of extubation and ICU discharge alive (as events of interest, both censored at 28 days) using the Kalbfleisch and Prentice method and a marginal Fine and Gray model, respectively, by treating death as a competing event. sHRs for cirrhotic versus non-cirrhotic patients were derived from the Fine and Gray model as the effect size. An sHR > 1 indicated a decrease in mechanical ventilation duration and length of ICU stay, whereas an sHR < 1 indicated an increase in mechanical ventilation and length of ICU stay. For all Fine and Gray models, the proportional hazards assumption for the subdistribution was assessed by examining the scaled Schoenfeld residuals plots. Statistical testing was conducted at the two-tailed α-level of 0.05. Data were analyzed using the SAS software version 9.4 (SAS Institute, Cary, NC, USA).

## 3. Results

### 3.1. At Inclusion

A total of 599 patients were included in the three randomized trials. Of those patients, 540 were eligible for this study. Forty patients with cirrhosis were identified. After matching, the cirrhotic group consisted of 39 patients, matched with 138 patients in the control group (Figure 1). Patient characteristics before matching are displayed in Table 1.

There were some imbalances between the cirrhosis group and the control group. We found a meaningful difference for shock at admission (84.6% vs. 68.1%), prone positioning (17.9% vs. 5.8%), heart failure (7.7% vs. 15.9%), and other characteristics, such as gastric reflux (10.3% vs. 3.6%) (Table 2).

There were no differences in terms of the number of tracheal aspirates per patient between the exposed and unexposed group (median: 16 (interquartile range: 10 to 20) vs. 16 (8 to 25); ASD = 4.9%). Tracheal pepsin measurements were available in 1181 aspirates. Alpha-amylase measurements were available in 1146 aspirates. 

### 3.2. Primary Outcome

No significant difference was observed between the two groups in terms of abundant gastric microaspiration (relative risk (RR): 0.31, 95%; CI: 0.75 to 1.10, *p* = 0.31) (Table 3). 

### 3.3. Secondary Outcomes

There was no significant difference in terms of the abundant microaspiration of oropharyngeal secretions, the duration of mechanical ventilation, the length of ICU stay and mortality. More cases of VAP were reported in cirrhotic patients ((Hazard Ratio: 1.91 (0.89 to 4.07) *p* = 0.094) compared to the non-cirrhotic patients, but the difference did not reach statistical significance (Figure 2).

### 3.4. Characteristics of VAP

VAP occurred in 11 patients in the cirrhosis group (28.2%): 4 qualified as early VAP and 7 as late-onset VAP. As for unexposed patients, VAP occurred in 22 patients (16.1%). 

## 4. Discussion

This study is, to our knowledge, the first to focus on cirrhosis as a possible risk factor for microaspiration in ventilated critically ill patients. There was no statistical difference between cirrhotic patients and the matched controls in terms of abundant gastric-content microaspiration. No statistical difference was observed in terms of the abundant microaspiration of oropharyngeal secretions, VAP cumulative incidence, the duration of mechanical ventilation, the length of ICU stay and mortality. 

The absence of a significant relationship between cirrhosis and microaspiration could be related to a lack of power as the number of cirrhotic patients was relatively small. Other well-known risk factors of microaspiration were frequently present in the two groups and could reduce the impact of cirrhosis on microaspiration. Other risk factors of microaspiration were not evenly distributed in the two groups: the cirrhosis group had a higher frequency of prone positioning (20% vs. 5.8% ASD = 43.3%) and pump-inhibitor treatment (67.5% vs. 56.5% ASD = 22.7%), whereas COPD was more frequent in the control group (23% vs. 15% ASD = 20.5). These differences could be partly responsible for our negative result. The absence of a difference between exposed and unexposed patients could also be explained by the high rate of gastric abundant microaspiration in all patients. In other words, risk factors for microaspiration, other than cirrhosis, were common in the two study groups. This suggests that cirrhosis might be a less important risk factor for the microaspiration of gastric contents in patients receiving invasive mechanical ventilation.

Several studies confirmed a higher susceptibility to bacterial infection in cirrhosis patients [16]. Cirrhosis is associated with a qualitative immunodepression with T-lymphocyte depletion and immunological alteration [17]. Nitric-oxide-induced vasoplegia and bacterial proliferation lead, in advanced cirrhosis, to bacterial translocation and low-grade inflammation [18]. This causes frequent transitory bacteremia and could trigger VAP. Impaired lung bacterial clearance, secondary to innate immune deficiency described in cirrhosis, could also be responsible for bacterial proliferation and subsequent VAP [19]. 

The absence of a significant relationship between cirrhosis and VAP in our study is probably related to the limited number of cirrhotic patients. Di Pasquale et al. identified cirrhosis as a risk factor for VAP [20]. They showed an increased mortality and poorer outcomes of VAP in the setting of liver disease. Cirrhosis was associated with a higher incidence of treatment failure and a higher occurrence of septic shock. This is corroborated by several other studies [21,22]. 

Several factors could influence the rate of gastric microaspiration in cirrhosis. Ascites and abdominal hypertension could be the main causes for the microaspiration of gastric contents. Unfortunately, due to the study design, no information could be provided on the incidence of ascites and abdominal hypertension in cirrhotic patients. A loss of consciousness associated with hepatic encephalopathy could also result in a higher risk of the microaspiration of gastric contents in cirrhotic patients [23]. Although no information was reported on hepatic encephalopathy in our study, few cases were presented with a coma. 

Our study had several strengths. First, microaspiration was diagnosed by the quantitative measurements of validated markers [4,14]. Second, our study included patients from three robust randomized controlled trials (RCTs) with all data prospectively collected. Thirdly, a VAP diagnosis was based on strict quantitative microbiological criteria, averting bias. 

Its retrospective design is one of the limitations. Another limitation was the relatively small number of cirrhotic patients (6.3% of the total study population). However, based on previous studies, cirrhotic patients represent about 10% of all ventilated patients [24]. 

## 5. Conclusions

To conclude, our results did not show an increased risk of gastric-content microaspiration in cirrhotic patients compared to the matched controls. Further studies should better investigate specific risk factors for pneumonia in cirrhosis. 

## Figures and Tables

**Figure 1 jcm-11-05994-f001:**
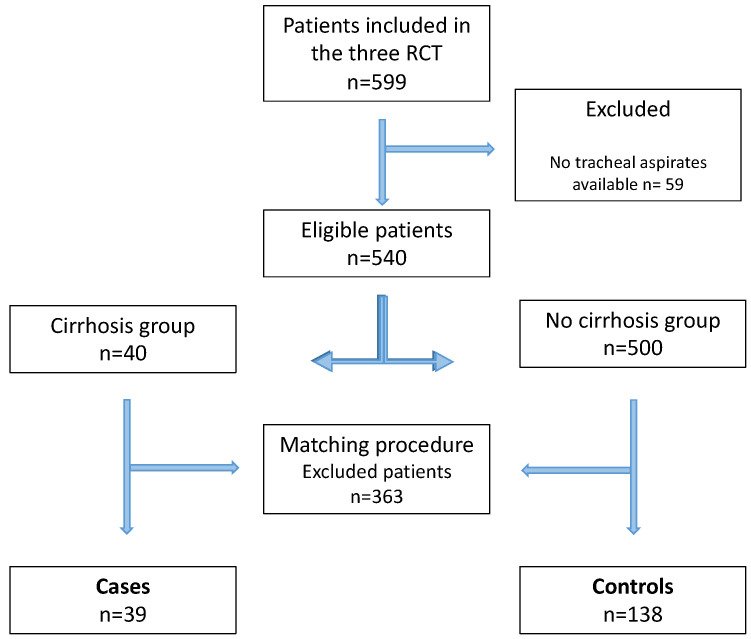
Flowchart of the study. RCT: randomized controlled trial.

**Figure 2 jcm-11-05994-f002:**
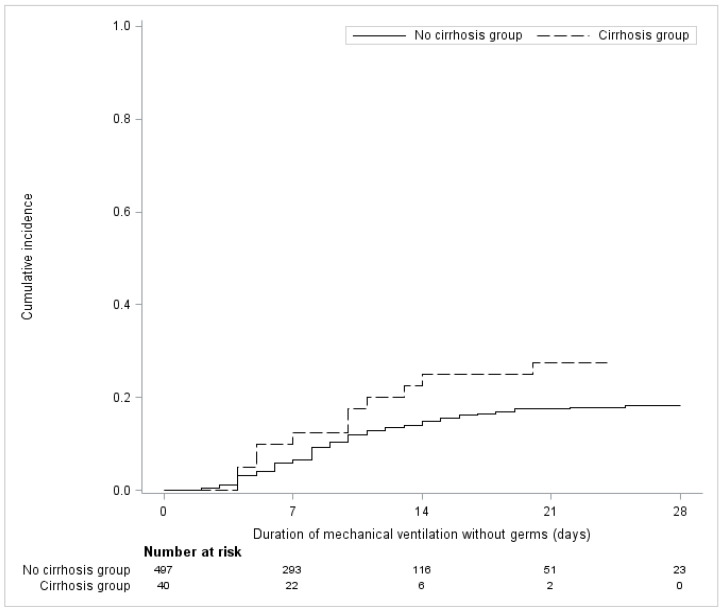
Cumulative incidence of VAP in patients with and without cirrhosis, after matching.

**Table 1 jcm-11-05994-t001:** Patient characteristics according to the group.

	No Cirrhosis Group *N* = 500	Cirrhosis Group *N* = 40	ASD (%)
Age (years), mean ± SD	63.0 ± 14.7	57.0 ± 12.4	44.0
Men	335 (67.0)	28 (70.0)	6.5
SAPS II, mean ± SD	51.6 ± 17.8	50.6 ± 17.7	5.7
Medical admission ^α^	352 (70.4)	30 (75.0)	15.5
Shock	331 (66.2)	34 (85.0)	44.9
Chronic Renal Failure	45 (9.0)	3 (7.5)	5.5
Gastric Reflux	26 (5.2)	4 (10.0)	18.2
Immunodepression	83 (16.6)	5 (12.5)	11.6
Diabetes	130 (26.0)	8 (20.0)	14.3
COPD	115 (23.0)	6 (15.0)	20.5
Heart failure	105 (21.0)	3 (7.5)	39.4
Enteral feeding	413 (82.6)	34 (85.0)	6.5
Stress ulcer prophylaxis	409 (81.8)	33 (82.5)	1.8
Pump proton inhibitor ^1^	281 (56.5)	27 (67.5)	22.7
Sedation	400 (80.0)	31 (77.5)	6.1
Prone positioning	29 (5.8)	8 (20.0)	43.3
Mean PEEP, median (IQR)	6 (5 to 8)	6 (5 to 10)	17.2
Mean GCS ^2^, median (IQR)	14 (7 to 15)	13 (6 to 15)	19.8
Number of tracheal aspirates ^3^ median (IQR)	14 (8 to 23)	16 (10 to 20)	18.4

Values are numbers (%) unless otherwise stated. ^α^: all patients were categorized as either medical or surgical admissions. ^1^: available on 537 patients (497 vs. 40). ^2^: available on 526 patients (489 vs. 37). ^3^ available on 536 patients (497 vs. 39). Abbreviations: ASD = absolute standardized difference; COPD = chronic obstructive pulmonary disease; IQR = interquartile range; GCS = Glasgow Coma Scale; PEEP = positive end-expiratory pressure; SAPS II = simplified acute physiology score II; SD = standard deviation.

**Table 2 jcm-11-05994-t002:** Patient characteristics according to the group, after matching.

	No Cirrhosis Group *N* = 138	Cirrhosis Group *N* = 39	ASD (%)
Age (years), mean ± SD	58.8 ± 10.9	57.8 ± 11.6	9.0
Men	104 (75.4)	28 (71.8)	8.1
SAPS II, mean ± SD	50.1 ± 16.4	50.3 ± 17.9	1.3
Medical admission ^α^	107 (77.5)	30 (76.9)	20.3
Shock	94 (68.1)	33 (84.6)	39.6
Chronic Renal Failure	8 (5.8)	3 (7.7)	7.6
Gastric Reflux	5 (3.6)	4 (10.3)	26.3
Immunodepression	27 (19.6)	5 (12.8)	18.4
Diabetes	29 (21.0)	8 (20.5)	1.2
COPD	27 (19.6)	6 (15.4)	11.0
Heart failure	22 (15.9)	3 (7.7)	25.8
Enteral feeding	114 (82.6)	33 (84.6)	5.4
Stress Ulcer Prophylaxis	116 (84.1)	32 (82.1)	5.4
Pump Proton Inhibitor	86 (62.3)	26 (66.7)	9.1
Sedation	113 (81.9)	30 (76.9)	12.3
Prone positioning	8 (5.8)	7 (17.9)	38.2
Mean PEEP, median (IQR)	6 (5 to 8)	6 (5 to 10)	13.0
Mean GCS ^1^, median (IQR)	13 (6 to 15)	13 (6 to 15)	13.3
Number of tracheal aspirates ^2^ median (IQR)	16 (8 to 25)	16 (10 to 20)	4.9

Values are numbers (%) unless otherwise stated. ^α^: all patients were categorized as either medical or surgical admission. ^1^: available on 173 patients (137 vs. 36). ^2^: available on 175 patients (137 vs. 38). Abbreviations: ASD = absolute standardized difference; COPD = chronic obstructive pulmonary disease; IQR = interquartile range; GCS = Glasgow Coma Scale; PEEP = positive end-expiratory pressure; SAPS II = simplified acute physiology score II; SD = standard deviation.

**Table 3 jcm-11-05994-t003:** Outcomes in patients with and without cirrhosis, after matching.

	No Cirrhosis Group *N* = 138	Cirrhosis Group *N* = 39	Effect Size	Value (95%CI)	*p*-Value
**Primary outcome**					
Abundant gastric micro aspiration	117/138 (84.8)	30/39 (76.9)	Relative risk	0.91 (0.75 to 1.10)	0.31
**Secondary outcomes**					
Abundant alpha-amylase measurements	95/137 (69.3)	23/38 (60.5)	Relative risk	0.87 (0.65 to 1.16)	0.34
VAP, number of events (cumulative incidence) at 28 days	22/137 (16.1)	11/39 (28.2)	Hazard ratio	1.91 (0.89 to 4.07)	0.094
Mechanical ventilation duration (days), median (IQR)	9 (5 to 25)	11 (5 to *not reach*)	Hazard ratio	0.75 (0.48 to 1.17)	0.20
length of ICU stay (days), median (IQR)	16 (9 to *not reach*)	17 (9 to *not reach*)	Hazard ratio	0.83 (0.50 to 1.37)	0.45
ICU mortality, number of events (cumulative incidence) at 28 days	34/138 (24.6)	13/39 (33.3)	Hazard ratio	1.41 (0.79 to 2.51)	0.24

Values are no./total no. (%) unless otherwise stated. Effect sizes are calculated for the cirrhosis group versus the non-cirrhosis group. Abbreviations: CI = confidence interval; ICU = intensive care unit; IQR = interquartile range; VAP = ventilator-associated pneumonia.

## Data Availability

Not applicable.

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
