# Peer review of "Are Cirrhotic Patients Receiving Invasive Mechanical Ventilation at Risk of Abundant Microaspiration"

_jcm, 2022, doi:10.3390/jcm11205994_

Round 1

Reviewer 1 Report

General comments:

This is a very interesting topic, providing useful information for the management of critically ill cirrhotic patients. While, as you mention, the study has several limitations regarding patient selection and design, the conclusion is well proven and documented. I believe that this article opens an exciting direction for scientific research.

Specific comments:

Line 88: please provide objective criteria for performing tracheal aspirates. Otherwise, the study is biased by the nurses’ experience.

Line 160: the phrase “and a few more” sounds unprofessional. Please use “such as:….” and then insert the reference to the table.

Lines 176-178: the instructions for article writing found in the template form were not deleted.

In tables 1 and 2, please explain the term “medical admission criteria”. Were there surgical patients?

Line 325: please revise the references, as reference 1 is missing from the list.

Author Response

Dear associate Editor,

Dear reviewer,

Thank you for the opportunity to submit a revised version of our manuscript. We revised the manuscript based on reviewers’ comments. Please find below our point by point response to reviewers’ comments.

Reviewer 1

This is a very interesting topic, providing useful information for the management of critically ill cirrhotic patients. While, as you mention, the study has several limitations regarding patient selection and design, the conclusion is well proven and documented. I believe that this article opens an exciting direction for scientific research. 

Comment

Line 88: please provide objective criteria for performing tracheal aspirates. Otherwise, the study is biased by the nurses’ experience. 

Response

We thank the reviewer for his comment, and agree with his suggestion. Tracheal aspirates are performed when bacteriologic samples are needed, when the patient’s experience discomfort and abundant respiratory secretion, or in the case of overflowing tracheal tube and desaturation. We modified the manuscript on line 88.

Comment

Line 160: the phrase “and a few more” sounds unprofessional. Please use “such as…” and then insert the reference to the table. 

Response

We changed this sentence according to the comment of the reviewer.

Comment

Lines 176-178: the instructions for article writing found in the template form were not deleted. 

Response

The instructions were deleted according to your comment.

Comment

In tables 1 and 2, please explain the term “medical admission criteria”. Were there surgical patients?

Response

The formulation <<medical admission criteria>> used in the initial version of the manuscript was confusing and actually referred to the category of admission.

Admissions were indeed categorized as medical or surgical. In our cohort, 382 patients were categorized as medical and 178 patients were categorized as surgical. We clarified this point in the footnotes of tables 1 and 2.

Comment

Line 325: please revise the references, as reference 1 is missing from the list. 

Response

We revised the references according to your comment.

Finally, we thank the reviewers and the Editor for their constructive comments.

Sincerely,

Dr Clementine Levy on behalf of the authors.

Reviewer 2 Report

This is a well drafted manuscript detailing the risk of microaspiration and ventilation-associated pneumonia (VAP) in patients with decompensated cirrhosis under mechanical ventilation in the ICU.

Authors assess risk in 39 patients with cirrhosis matched with 138 controls and reported increased incidence of VAP in the cirrhosis arm which was not statistically significant. The findings are interesting, and my suggestions follows.

MAJOR 

1.       Line 174-175: 4 and 9 does not add up to 11. Please clarify.

2.       Line 176: do you mean 13 and 9? Please simplify/clarify.

3.       Lines 176-178: Is the line supposed to be in the manuscript? Please check and correct.

4.       Lines 252-256: I suggest moving lines 252-256 into Line 258  i.e.,

“This study is to our knowledge the first to focus on cirrhosis as a possible risk factor for microaspiration in ventilated critically ill patients. There was no statistical difference between cirrhotic patients and matched controls in terms of abundant gastric content microaspiration. No statistical difference was observed in terms of abundant microaspiration of oropharyngeal secretions, VAP cumulative incidence, duration of mechanical ventilation, ICU length of stay and mortality.”

Ideally, the unique aim of a study should be the opening line of the discussion. It reads better that way.

MINOR

i.                     Abbreviations should be spelled out the first time they appear e.g., SAP II in abstract.

ii.                   Line 81: “We used…” should be corrected to reflect the fact that the study is retrospective.

iii.                 Line 278: “…responsible for…”.

iv.                 Lines 288-290: Please provide reference on the relationship between HE and microaspiration.

v.                   Line 273: “T-lymphocyte” not “lymphocyte-T”.

Author Response

Dear associate Editor,

Dear reviewer,

Thank you for the opportunity to submit a revised version of our manuscript. We revised the manuscript based on reviewers’ comments. Please find below our point by point response to reviewers’ comments.

Reviewer 2

This is a well drafted manuscript detailing the risk of microaspiration and ventilation-associated pneumonia (VAP) in patients with decompensated cirrhosis under mechanical ventilation in the ICU.

Authors assess risk in 39 patients with cirrhosis matched with 138 controls and reported increased incidence of VAP in the cirrhosis arm which was not statistically significant. The findings are interesting, and my suggestions follows.

MAJOR suggestions

Comment

Line 174-175: 4 and 9 does not add up to 11. Please clarify.

Response

This was a typo that we fixed in the revised version. We report 11 ventilator associated pneumonia (VAP): 4 early and 7 late onset VAP.

Comment

Line 176: do you mean 13 and 9? Please simplify/clarify.

Response

We simplified the manuscript according to your comment.

Comment

Lines 176-178: Is the line supposed to be in the manuscript? Please check and correct.

Response

The instructions from the template were deleted.

Comment

Lines 252-256: I suggest moving lines 252-256 into Line 258  i.e.,

“This study is to our knowledge the first to focus on cirrhosis as a possible risk factor for microaspiration in ventilated critically ill patients. There was no statistical difference between cirrhotic patients and matched controls in terms of abundant gastric content microaspiration. No statistical difference was observed in terms of abundant microaspiration of oropharyngeal secretions, VAP cumulative incidence, duration of mechanical ventilation, ICU length of stay and mortality.”

Response

We thank the reviewer for this suggestion and modified the manuscript accordingly.

MINOR suggestions

Comment

Abbreviations should be spelled out the first time they appear e.g., SAP II in abstract.

Response

We spelled out every abbreviation in the manuscript.

Comment

Line 81: “We used…” should be corrected to reflect the fact that the study is retrospective

Response

We thank the reviewer for this suggestion and modified the manuscript to reflect the retrospective design. Accordingly we state in the revised version that << Polyvinyl chloride tracheal tubes were used in all patients included in the trials. >>

Comment

Line 278: “…responsible for…”

Response

We modified the manuscript according to your suggestion.

Comment

Lines 288-290: Please provide reference on the relationship between HE and microaspiration.

Response

Hepatic encephalopathy especially in critically ill cirrhotic is associated with coma and severe neurological manifestation which in turn led to higher frequency of microaspiration (Metheny NA, Clouse RE, Chang YH, Stewart BJ, Oliver DA, Kollef MH. Tracheobronchial aspiration of gastric contents in critically ill tube-fed patients: frequency, outcomes, and risk factors. Crit Care Med. 2006 Apr;34(4):1007-15.)

We added this reference in the revised version.

Comment

Line 273: “T-lymphocyte” not “lymphocyte-T”.

Response

We modified the manuscript according to your suggestion.

Finally, we thank the reviewers and the Editor for their constructive comments.

Sincerely,

Dr Clementine Levy on behalf of the authors

Reviewer 3 Report

Sir/Madam,

this paper is well written; it is informative; discussion of results is balanced. On the whole, collected data are of some interest but, honestly, the authors do not miss to underline that this study has limits (relatively small number of recruited cirrhotic patients, lack of some key clinical information about these cirrhotics: presence of ascites? previous encephalopathy? etc.) As stated, the authors are honest enough not to hide these limitations.

Anyway, conclusions are firmly supported by the data provided.

Author Response

Dear reviewer, 

We thank you sincerely for your kind reviewing and stay at your disposal for any further question regarding this manuscript.

Sincerely, 

Dr Levy on behalf of the authors

Reviewer 4 Report

The article is fascinating because of the results the Authors have shown. Liver cirrhosis is a disease that is related to an increased infection rate and this paper shows quite different results. The Authors do realize this fact and try to find the source of this fact in the discussion chapter. I have to agree that perhaps the analyzed group size was/is a reason for the results. The methodology seems to be logical and proper and still the results are surprising. I thought that maybe there could be an attempt to increase the sample size. Still, among the studied patients, only 40 were cirrhotic and 39 were selected for analysis which makes it impossible to increase the study group size to enlarge available data.

I found only small issues, mostly all technical such as double spaces (for example in lines 43, 63), lack of space between a word and opening parenthesis (for example in line 128), too big indent in line 129, and so on. These issues are not severe and require only minor editorial work. 

Author Response

Dear reviewer, 

We thank you kindly for you helful comments.   We made the grammar editing as requested by the reviewer in the revised version.   Sincerely,    Dr Levy on behalf of the authors